# PTEN Function at the Interface between Cancer and Tumor Microenvironment: Implications for Response to Immunotherapy

**DOI:** 10.3390/ijms21155337

**Published:** 2020-07-27

**Authors:** Fabiana Conciatori, Chiara Bazzichetto, Italia Falcone, Ludovica Ciuffreda, Gianluigi Ferretti, Sabrina Vari, Virginia Ferraresi, Francesco Cognetti, Michele Milella

**Affiliations:** 1Medical Oncology 1, IRCCS Regina Elena National Cancer Institute, 00144 Rome, Italy; fabiana.conciatori@ifo.gov.it (F.C.); italia.falcone@ifo.gov.it (I.F.); gianluigi.ferretti@ifo.gov.it (G.F.); sabrina.vari@ifo.gov.it (S.V.); virginia.ferraresi@ifo.gov.it (V.F.); francesco.cognetti@ifo.gov.it (F.C.); 2SAFU, Department of Research, Advanced Diagnostics, and Technological Innovation, IRCCS Regina Elena National Cancer Institute, 00144 Rome, Italy; ludovica.ciuffreda@ifo.gov.it; 3Division of Oncology, University of Verona, 37126 Verona, Italy; michele.milella@univr.it

**Keywords:** PTEN, cancer, immune cells, immunoevasion, immunotherapy

## Abstract

Mounting preclinical and clinical evidence indicates that rewiring the host immune system in favor of an antitumor microenvironment achieves remarkable clinical efficacy in the treatment of many hematological and solid cancer patients. Nevertheless, despite the promising development of many new and interesting therapeutic strategies, many of these still fail from a clinical point of view, probably due to the lack of prognostic and predictive biomarkers. In that respect, several data shed new light on the role of the tumor suppressor phosphatase and tensin homolog on chromosome 10 (PTEN) in affecting the composition and function of the tumor microenvironment (TME) as well as resistance/sensitivity to immunotherapy. In this review, we summarize current knowledge on PTEN functions in different TME compartments (immune and stromal cells) and how they can modulate sensitivity/resistance to different immunological manipulations and ultimately influence clinical response to cancer immunotherapy.

## 1. Introduction

Tumorigenesis is a genetic/epigenetic process driven by oncogene activation and tumor suppressor gene inactivation [1]. Phosphatase and tensin homolog on chromosome 10 (*PTEN*) is one of the tumor suppressors most frequently inactivated in human cancer, due to genetic alterations or transcriptional/post-transcriptional inhibition; moreover, even a partial loss of its function (haploinsufficiency) may cause neoplastic transformation [2,3,4]. Hence, we will refer to either genetic mutations or protein lost as “PTEN-loss” [5]. The PTEN protein mainly acts as a lipid phosphatase, which converts phosphatidylinositol 3, 4, 5 trisphosphate (PIP3) into phosphatidylinositol 4, 5-bisphosphate (PIP2), counteracting the activity of the phosphoinositide 3-kinase (PI3K) and resulting in the inhibition of cell proliferation, survival and migration [2,6].

The regulation of PTEN expression and function in cancer cells is extremely complex and the recognition of its role as a predictive/prognostic biomarker is hampered by the lack of unequivocal methods to ascertain whether the protein is non-functional, or present [4,6,7]. A further, emerging level of complexity is related to the possibility that the loss of PTEN function may directly or indirectly influence not only cancer cell behavior, but also the tumor microenvironment (TME) and immune-infiltrate composition and function [8]. Since the interaction between cancer and stromal/immune cells may result in a tumor-permissive or non-permissive TME, PTEN activity is in a crucial position to control the overall effects of such interactions.

Immune escape represents one of the hallmarks of cancer and can be determined by a combination of relatively low cancer cell immunogenicity and tumor-dependent immunosuppression [9]. The recognition of the role of immune checkpoints, particularly cytotoxic T-lymphocyte antigen (CTLA)-4 and programmed cell death (PD)-1, has shed new light on the mechanisms of negative regulation of the immune system and opened a new era in the clinical application of immunotherapy in cancer [10,11]. In such a complex scenario, the continuous bidirectional interactions between cancer cells and infiltrating immune/inflammatory cells are dictated, at least in part, by the genetic background of the different cancerous and non-cancerous components. Such interactions, in turn, may crucially determine the sensitivity or resistance to immunotherapeutic approaches and provide novel targets for an effective treatment.

In this review, we focus on the possible implications of PTEN expression and function within different TME compartments (non-cancer cells and soluble factors), in order to better understand potential mechanisms underlying immunotherapy resistance.

## 2. Current Status of Clinical Cancer Immunotherapy

Immunotherapy is approved for the treatment of an ever-expanding array of advanced malignancies of different histological origin [12]. According to their mode of action, immunotherapies can be categorized into strategies that stimulate the effector machinery or neutralize immunosuppression mechanisms (summarized in Table 1) [13].

### 2.1. Stimulation of the Effector Machinery

Conventional preventive vaccines are used in healthy individuals to prevent infectious diseases. Conversely, therapeutic cancer vaccines are used in patients already affected by the disease, to prevent cancer progression, by boosting the action of the immune system [13]. Despite the failure of many cancer vaccine trials, two types of cancer vaccines are currently approved, an autologous dendritic cell (DC)-based vaccine for prostate cancer (Sipuleucel-T) and an oncolytic herpesvirus-based vaccine for metastatic melanoma (T-VEC) (revised in [15,16]). Adoptive T-cell transfer represents an alternative therapeutic strategy which employs ex vivo-activated or genetically engineered immune system cells, such as tumor-infiltrating lymphocytes (TILs) and chimeric antigen receptor T-cells (CAR-T) [34]. While the potential of TILs is currently limited to melanoma, CAR-T have shown great promise, particularly in hematological malignancies; conversely, their use to treat solid tumors has been limited so far, possibly due to immunosuppressive interactions between the TME and CAR-T [19,20,35,36].

### 2.2. Counteract the Immunosuppressive Mechanisms

The recognition of the biological role and therapeutic potential of immune checkpoint blockade (through CTLA-4 and PD-1 targeting) has revolutionized our approach to cancer immunotherapy, as also highlighted by the 2018 Nobel Prize award [11,37]. Many different monoclonal antibodies (mAb) targeting CTLA-4, PD-1, or PD ligand (PD-L)1, such as ipilumumab and nivolumab, pembrolizumab, atezolizumab, durvalumab and avelumab, have been approved for the treatment of specific cancers [25,38,39]. Although at higher risk of important immune-related toxicity, the combined inhibition of CTLA-4 and PD-1 signaling, using ipilimumab and nivolumab, has shown increased clinical efficacy in melanoma, kidney and lung cancer. Moreover, a second generation of immune checkpoint inhibitors (revised in [40]) is underway: among others, lymphocyte activation gene (LAG)-3, T-cell immunoglobulin mucin (TIM)-3, and T cell immunoglobulin and ITIM domain (TIGIT) represent the most promising cancer therapeutic targets, expressed by natural killer (NK), CD4^+^ and CD8^+^ T cells, as redundant co-inhibitory receptors [26,41,42,43]. Another combinatorial therapeutic strategy aims at both blocking inhibitory signals and stimulating activation/co-stimulatory signals [44]. For example, preclinical studies reported encouraging data for the simultaneous engagement of the co-stimulatory receptor 4-1BB (also known as CD137 or TNFSF9) in combination with an anti-PD-1 blockade [45].

## 3. PTEN in Immunoevasion

Mounting evidence suggests that PI3K signaling may influence the composition and functionality of the TME, thereby modulating immune response in cancer [46]. In particular, PTEN expression (or lack thereof) in cancer cells attracts different immune cell populations to the TME; on the other hand, PTEN function in immune cells regulates their activation status [8]. As schematically depicted in Figure 1, the overall effect of PTEN loss of function in different cellular compartments shifts the balance towards an immunosuppressive TME [47,48,49]. Here, we give a comprehensive overview of the potential role of PTEN in the regulation of the immunosuppressive aspects of the TME (Figure 1).

### 3.1. PTEN Role in Immune Cells

DCs are professional antigen-presenting cells (APCs) implicated in adaptive immunity, through foreign antigens processing and the subsequent major histocompatibility complex (MHC)-dependent presentation. This mechanism results in the stimulation of naïve T cells and cytotoxic effector cells (i.e., macrophages, NKs) function [50,51]. Failure to effectively present antigens or functional deficiencies in infiltrating DCs contributes to immune suppression [51]. The PI3K pathway plays a key role in DC functions. Indeed, DCs derived from PI3K*γ*^−/−^ mice display reduced ability to migrate in response to chemoattractants; similarly, antigen-loaded DCs also show decreased ability to move to lymph nodes [52]. Higher PTEN levels were observed in DCs of elderly, as compared to young, subjects, resulting in reduced AKT activation, antigen-uptake, and DC migration [53]. Based on the above evidence, targeting PTEN in DC-based cancer vaccines could represent a promising approach in immunotherapy. The advantageous effects exerted by the use of PTEN-silenced DCs were related not only to increased DC survival and CCR7-dependent migration, but also to enhanced CD8^+^ numbers [54]. Pan and collaborators demonstrated that PTEN cooperates in the negative regulation of Dectin-1 and FcεRI γ-chain (FcRγ)-mediated signaling. FcRγ represents an adapter of immunoreceptor tyrosine-based activation motif (ITAM) on DCs and Dectin-1 is a receptor containing an ITAM-like domain involved in DC function in the immune response. Though mostly implicated in antifungal response and leucocyte recognition, the authors underline the importance of PTEN-mediated negative regulation and hypothesize that PTEN targeting may be used as a strategy for the development of new cancer therapy approaches [55].

By modulating antigen receptor expression and cytokine release, PI3K signaling represents a central hub in the regulation of normal T cells differentiation into either cytotoxic CD8^+^ or helper CD4^+^. As a consequence of PIP3 production and mammalian target of rapamycin (mTOR) complex 2 phosphorylation, the transcription factor forkhead box (FOX)O translocates from the nucleus to the cytoplasm, resulting in the decreased expression of genes involved in cell cycle arrest and T cell differentiation [56]. PTEN-loss results in a persistent FOXO inactivation and resistance to apoptotic cytokine-mediated signaling, as often detected in hemangiomas and leukemias [57]. Moreover, PTEN heterozygous (PTEN^+/−^) mice are deficient in Fas-mediated apoptosis: as the binding of Fas (also known as CD95) to its ligand (FasL) is the main mechanism by which CD8^+^ cytotoxic T-cells kill non-self-cancer cells, PTEN haploinsufficiency results in lymphoid hyperplasia and tumor growth [58,59]. In CD4^+^ helper T cells, on the other hand, PTEN expression patterns result in opposite effects, according to the timing of PTEN gene inactivation. Indeed, thymocyte-specific PTEN deletion causes lymphomas and autoimmunity, whereas activated T cells hyper-proliferate and over-express cytokines in the presence of PTEN-loss [60,61].

PI3K signaling also regulates CD4^+^ regulatory T cell (Treg) functions. After T cell receptor engagement and cell activation, PTEN is down-regulated, and this results in interleukin (IL)-2 dependent PI3K signaling and Tregs expansion. On the other hand, IL-2 is able to promote PI3K pathway activation, despite high levels of PTEN, leading to T cell anergy. Overall, the loss of PTEN function cooperates with CD25 stimulation by IL-2 to promote Treg proliferation in a clinical setting [62,63]. PTEN expression may also inhibit T cell response through intercellular binding of PD-1/neurophilin-1 (Nrp-1) and PDL-1/semaphorin-4 (Sema4a) on Tregs and effector cells surface, respectively. As demonstrated by Francisco and coworkers, PD-1 upregulation attenuates PI3K activation during Tregs maturation [64]. It has also been demonstrated that the Sema4a/Nrp1 axis attenuates Tregs-mediated antitumor immune response: the binding of the ligand Sema4a, expressed on immune cells, prevents AKT activation through PTEN-dependent FOXO3a nuclear localization [65]. Another immunosuppressive mechanism, potentially influenced by PTEN, is mediated by the indoleamine 2, 3-dioxigenase (IDO) enzyme, which prevents tryptophan-mediated immunological stimulus and promotes kynurenine-dependent immune effector cells disruption. Intricate feedback signals exist, by which IDO cooperates with PD-1 to promote an immunosuppressive Tregs phenotype. IDO expression in DCs and APCs activates PTEN signaling in Tregs, thus blocking the PI3K cascade and increasing the activity of FOXO1 and FOXO3a, which in turn upregulate PD-1 and PTEN [66]. PTEN inactivation after PD-1 blockade abrogates FOXO-dependent immunosuppression and results in cancer regression [67,68,69].

CD56^+^/CD3^−^ large granular NK cells recognize and destroy both infected and transformed cells. A variety of activating stimuli is necessary to fully activate the NK cytotoxicity: once activated, NK cells release perforin, granzymes, and antitumoral immune response-promoting cytokines (e.g., IFN-γ and tumor necrosis factor (TNF)-α) and induce apoptosis of targeted cells, through FasL/Fas or TNF-related apoptosis-inducing ligand (TRAIL)/TRAIL receptor interactions [70]. Consistently, a marked depletion of their number/activity correlates with increased risk of developing cancer and cancer progression [60]. Immature CD56^bright^ NK cells express higher levels of PTEN, as compared to cytotoxic, CD56^dim^ cells, suggesting a specific role of PTEN in NK cytotoxicity. In this context, PTEN disrupts the immunological synapses between NKs and targeted cells, by decreasing actin accumulation, polarization of microtubules and cytolytic granule mobilization [71]. Another group demonstrated that PTEN is also involved in NK activity by affecting their trafficking and localization in vivo. Indeed, the authors showed a decreased percentage of mature NK cells in peripheral compartments in PTEN-knock out (ko) NK mice [72]. Finally, PTEN is a central player in NK cells activation, through inhibition of the PIP3-mediated signaling cascade (extensively revised in [73]).

Myeloid-derived suppressor cells (MDSCs) are among the key actors involved in mediating tumor escape mechanisms in the TME, due to their specific ability to abolish T cell and NK functions [74,75]. It has been demonstrated that a complex network of micro-RNAs (miRs) regulates MDSCs activation and functions in the TME, thus opening new scenarios for novel immunotherapy approaches [76]. Tumor-derived transforming growth factor (TGF)-*β*1 upregulates miR-494, which in turn regulates MDSCs activity and localization. *PTEN* is a direct target of miR-494 and its downregulation, with the consequent activation of PI3K pathway, is involved not only in C-X-C chemokine receptor (CXCR)4-mediated MDSCs chemotaxis towards tumors, but also in metastases formation due to upregulation of metalloproteinases (MMP) (e.g., MMP2, MMP13) [77]. Moreover, TGF-*β* is also involved in modulating MDSCs expansion through the inhibition of PTEN and SH-2 containing inositol 5′ polyphosphatase (SHIP)1. TGF-*β*-treated bone marrow-MDSCs display lower levels of *PTEN* and *SHIP-1*, regulated by miR-21 and miR-155 respectively, and higher levels of signal transducer and activator of transcription (STAT)3: all these perturbations result in an increased number of MDSCs [78]. Several authors also elucidated the importance of the *PTEN* mRNA 3′UTR as the “seed sequence” recognized by specific miRs, in the regulation of MDSC proliferation and activity [79,80]. Mei and collaborators demonstrated that, among the soluble factors produced by tumor cells, the granulocyte–macrophage colony-stimulating factor is the main inducer of miR-200c. This miR is involved in the upregulation of MDSC-immunosuppressive functions and proliferation, by directly targeting the 3′UTR of both *PTEN* and friend of Gata 2 (*FOG2*). Inhibition of *PTEN* and *FOG2*, in turn, activates PI3K leading to the MDSC differentiation, and STAT3, involved in the production of pro-inflammatory, pro-proliferative and anti-apoptotic factors [79]. Interestingly, tumor-derived exosomes may exert similar effects on PTEN regulation in MDSCs. Indeed, miR-107 delivered by gastric cancer-derived exosomes exerts its activity by inhibiting the functions of *PTEN* and *Dicer1*, through the binding to the 3′UTR region, thus resulting in MDSC proliferation and acquisition of ARG-1-mediated suppressive function [80,81]. Similarly, glioma-derived exosomes act as shuttles for miR-21: the consequent inhibition of *PTEN* and activation of STAT3 promote bone marrow-derived MDSC proliferation and differentiation. Moreover, PTEN-silenced MDSCs upregulate the production of the immunosuppressive cytokine IL-10 [81].

Macrophages are conventionally classified into two categories according to the membrane receptors and their functions: M1 and M2 (or tumor-associated macrophages—TAM), endowed with tumor-suppressor and tumor-promoting properties, respectively. M2 macrophages are the most represented leukocytes in cancer stroma, representing up to 50% of the immune cells in the TME [46]. Li and colleagues showed that PTEN-silencing in macrophages causes an increased release of C-C motif chemokine ligand (CCL)-2 and vascular endothelial growth factor (VEGF)-A, promoting the M2 phenotype switch [82]. Moreover, *PTEN* deletion results in the expression of specific M2 markers, such as the immunomodulating protein arginase I [83]. By enhancing arginase I, M2 macrophages reduce T cell proliferation, hence promoting T cell anergy [84]. A recent paper demonstrated that PTEN expression, regulated by miR networks, can modulate macrophage polarization. Supernatants derived from human glioblastoma cells upregulated miR-32, which in turn interacts and suppresses PTEN in an in vitro model of human monocytes, thus promoting the M2 phenotype and resulting in an enhanced cell proliferation [85]. miR-21 reduces the expression of *PTEN* and its upstream positive regulator miR-200c in primary macrophages, promoting their differentiation into M2 [86]. Exosomes derived from pancreatic cancer cells promote hypoxia-inducible factors (HIF)-1*α*/HIF-2*α*-dependent M2 phenotype through PTEN regulation: miR-301a-3p converts stromal macrophages into M2 macrophages and promotes lung metastases formation by blocking PTEN transcription [87]. Hypoxic lung tumors release miR-103a which blocks *PTEN* activity: the consequent PI3K/AKT activation induces M2 polarization, with high ability to migrate and regulate angiogenesis [88]. In a very intricate bidirectional crosstalk between cancer cells and the surrounding TME, under hypoxic conditions, epithelial ovarian cancer TAMs release miR-223, which in turn regulates *PTEN*, promoting ovarian cancer cell proliferation and drug resistance, through PI3K activation [89].

### 3.2. PTEN Role in Stromal Cells

PTEN genetic inactivation in fibroblasts has extensively been studied in breast cancer [90,91,92]. Trimboli and coworkers demonstrated that *PTEN*-loss in fibroblasts results in an increased incidence of HER2-driven breast tumor, innate immune cells infiltration and VEGF-dependent angiogenesis. Indeed, *PTEN*-loss results in Ets2 phosphorylation and inactivation, thus allowing for the activation of a specific transcriptional program, associated with a more aggressive tumor behavior [90]. Moreover, the lack of *PTEN* in mammary stromal fibroblasts is associated with modulation of both fibroblasts and other surrounding TME cells. *PTEN*-loss modifies the oncogenic secretome through the downregulation of miR-320, hence reprogramming both endothelial and epithelial cells towards a more malignant phenotype [91]. The specific deletion of *PTEN* in murine fibroblasts promotes collagen deposition and parallel alignment of cellular matrix (a feature observed also in patients with breast cancer) even in the absence of cancer cells. Furthermore, *PTEN*-loss in fibroblasts fosters cancer-associated fibroblasts-like behaviors, such as upregulation of *α*-smooth muscle actin and MMP activity [92]. Mechanisms of PTEN inactivation have been investigated in pancreatic ductal adenocarcinoma (PDAC), using high throughput techniques. A recent study showed a surprisingly high percentage (40%) of PTEN-loss (due to partial deletion of chromosome 10) in stromal cells, while PTEN loss of heterozygosity was observed in 46.6% of tumors and juxta-tumoral stroma in PDAC patients. In addition, miR-21 overexpression in juxta-tumoral stroma cooperates with PTEN inactivation, even in the absence of PTEN mutations. These observations correlate with aggressive tumor features and worse prognosis [93]. PTEN-loss in PDAC fibroblasts is usually associated with the ablation of the smoothened (Smo) gene, resulting in PTEN degradation by the E3 ubiquitin ligase RNF5: proteasome-dependent PTEN inactivation leads to increased proliferation and reduced overall survival (OS) [94].

As widely recognized, PTEN also modulates angiogenesis [95]. Tian and collaborators demonstrated that transfection of HepG2 cells with wild type (wt) PTEN suppresses the expression of VEGF in a HIF-1-dependent manner. Transfection with a PTEN construct lacking the C2 phosphatase domain also resulted in downregulation of both VEGF and angiogenesis in vitro and in vivo, albeit to a lesser extent as compared with the complete PTEN construct. This evidence suggests a possible cooperation between phosphatase-dependent and -independent PTEN functions in regulating angiogenesis [96]. The tight link between PI3K/PTEN activation and VEGF was also investigated by our group: indeed, we demonstrated that the hyperactivation of PI3K pathway leads to HIF-1/2 translation and subsequent expression of VEGF [97]. Dong and collaborators showed that PTEN-defective melanoma cell lines express several cytokines, including VEGF, and their expression is transcriptionally inhibited by the PI3K inhibitor LY294002 [98]. A similar correlation was identified also for another pro-angiogenic factor: de la Iglesia and co-workers demonstrated that STAT3-dependent IL-8 expression occurs only in PTEN-loss glioblastoma contexts [99].

### 3.3. Tumor PTEN Affects Immune Infiltrate

A wealth of evidence suggests a central role of tumor PTEN status in modulating the TME immune infiltrate and cancer cells/TME interactions in different tumor histotypes (Figure 2).

In mice and human lung squamous cell carcinoma, the TME of PTEN-loss samples is characterized by the specific accumulation of tumor-associated neutrophils and Tregs, involved in different processes such as angiogenesis and immunosuppression. Moreover, low levels of TAMs, NKs, T and B cells during tumor progression were observed according to increasing tumor burden [100]. In vitro experiments in *PTEN*-silenced triple negative breast cancer (TNBC) show that low levels of infiltrating CD4^+^ and CD8^+^ T cells are due to the induction of apoptotic mechanisms associated with PD-L1 expression and PTEN-loss [101]. In many instances, tumor genetic background, *PTEN*-loss in particular, shapes an immunosuppressive TME through the production of specific soluble factors, which in turn modify stromal/immune cells infiltration. Data from our and other groups suggest that PTEN-loss in prostate epithelium, glioblastoma and colorectal cancer (CRC) cells promotes a selective increase in IL-8 expression [99,102,103,104,105]. IL-8 production, in turn, correlates with an immunosuppressive, myeloid-enriched, TME and unequivocally with an adverse cancer prognosis, particularly for patients undergoing immunotherapy. In CRC patients, serum IL-8 levels are associated with activation of a gene expression program which enforces a monocyte-/macrophage-like phenotype in CD4^+^ T cells [106]. FOXP3 CD4^+^ Tregs markedly express the IL-8 receptor CXCR1, in order to respond to tumor-derived IL-8 and migrate in cancer tissue, and CD4^+^ T cells produce IL-8 themselves [107,108]. On the other hand, CXCR1 expression is downregulated on CD8^+^ T cells after antigen presentation, thereby reducing the number of cytotoxic T cells [109]. Interestingly, a recent paper reported that mesenchymal stem cells release IL-8, which activates c-Myc via STAT3 and mTOR signaling in gastric cancer cells, hence resulting in membrane PD-L1 overexpression and blockade of cytotoxic effects of CD8^+^ T cells. In this complex crosstalk between different cell population, neutralizing IL-8 could enhance the efficacy of PD-L1 antibodies [110]. In a cohort of 168 resected CRC patients, our group recently demonstrated that PTEN-loss in cancer cells significantly correlates with high levels of tumor-derived IL-8 and low levels of IL-8^+^ infiltrating mononuclear cells (Conciatori 2020, unpublished data, [103]).

Toso and coworkers showed that in *PTEN*-loss prostate cancer activation of JAK2 and phosphorylation of STAT3 induce the production of specific chemokines, which recruit an increased number of infiltrating MDSCs [111]. Similarly, in PTEN-loss melanoma xenograft models the upregulated production of soluble factors (e.g., CCL2, VEGF) recruits and/or regulates the function of suppressing immune cells (e.g., immature DCs, MDSCs), thus promoting an immunosuppressive TME. In metastatic melanoma patients, nanostring-based analysis of the TME inflammatory cells showed an 83% reduction in inflammation-related genes associated with the *PTEN*-loss status of cancer cells [112]. In a *HRAS^G12V^*/*PTEN^−/−^* follicular thyroid carcinoma mouse model immunosuppressive Tregs and M2 macrophages were significantly upregulated. Moreover, cell lines isolated by *HRAS^G12V^*/*PTEN^−/−^* express higher levels of chemotactic factors for MDSCs, T cells and macrophages as compared to *BRAF^V600E^*/*PTEN^−/−^* tumors, highlighting the tight association between genetic background and tumor immune infiltrate [113]. Tumor genetic background influences immune/inflammatory infiltrate into the TME not only in the primary tumor, but also in metastatic lesions, as demonstrated by Vidotto et al. Indeed, the authors showed that in *PTEN*-loss prostate cancer, higher levels of Tregs were observed in liver metastases as compared to the primary lesion, and high levels of CD8^+^ cells were present in bone metastases [114].

## 4. PTEN and Immunotherapy

### 4.1. PTEN and Conventional Immune Checkpoint Inhibitors

In recent years, many steps have been taken to increase the number of patients who may benefit from immunotherapy by understanding the mechanisms behind the primary and acquired resistance [115]. PTEN expression and function may contribute to the regulation of immune homeostasis and sensitivity/resistance to immune checkpoint inhibition-based therapeutic strategies. An inverse correlation between PD-L1 and PTEN expression has been demonstrated, both in vitro, using *PTEN*-silenced cellular models, and in vivo. In a series of 404 CRC patients PTEN-loss and PD-L1 overexpression significantly correlated with poor OS and higher tumor, node, metastasis (TNM) stage [116]. Conversely, treatment with an anti-PD-L1 antibody decreases tumor growth by upregulating PTEN levels in both pancreatic cancer tissue and liver metastases in mice models [117]. In vitro experiments show that PTEN-loss or PI3K pathway activation increase PD-L1 levels in glioma cell lines, resulting in decreased sensitivity to T cell-mediated killing [118]. Moreover, Waldron and colleagues correlated PTEN status with autologous T cell apoptosis in glioblastoma patients. PTEN-loss in glioblastoma cells fosters CD3^+^ T cells apoptosis upon cell-cell contact and such effect can be reverted by pre-treatment with PI3K/AKT pathway inhibitors, suggesting a possible therapeutic strategy encompassing PI3K/AKT-targeted inhibitors in combination with immunotherapy [119]. Along these lines, a recent case report analyzed drug response of a metastatic non-small-cell lung cancer (NSCLC) patient, whose tumor lacked PTEN expression: the authors described immunotherapy-resistance despite high levels of PD-L1 expression, while the tumor responded to rapamycin analogs [120]. Consistently, treatment with either AKT or mTOR inhibitor resulted in decreased PD-L1 expression in PTEN-loss TNBC: PD-L1 reduction, and the consequent increased T cell proliferation and apoptosis, could enhance antitumor immunity [101].

Despite the presence of high levels of PD-L1 in PTEN-loss tumor, a paradoxical resistance to PD-1 blockade often occurs in PTEN-loss contexts [121]. A recent study in a cohort of glioblastoma patients treated with an anti-PD-1 mAb (nivolumab or pembrolizumab) upon recurrence revealed that a greater percentage of *PTEN* mutations is observed in non-responders (72%), as compared to responders (23%) and that *PTEN*-loss is also associated with a specific immunosuppressive signature. In particular, in *PTEN*-loss patients the immunosuppressive TME is characterized by both high levels of CD68^+^/HLA^-^/DR^-^ macrophages and the absence of immune infiltration [122]. Similar results were recently reported by Chen and coworkers: they analyzed all somatic genetic mutational status of 113 NSCLC patients in response to CTLA-4/PD-1 blockade and detected *PTEN* mutations exclusively in non-responders, albeit at a low frequency (6.8%) [123]. Failure to respond to the immune checkpoint blockade in the presence of *PTEN*-loss was widely described also in metastatic melanoma. A recent analysis conducted in metastatic melanoma patients treated with an anti-CTLA-4 followed by an anti-PD-1 mAb, identified recurrent loss of different tumor suppressor genes, including *PTEN*, in double non-responder patients. Among the tumor suppressor genes analyzed, *PTEN*-loss appeared to drive resistance to CTLA-4 blockade in particular [124]. However, published data conflict in regard to the correlation between PTEN status and anti-CTLA-4 response; in that respect, it is essential to keep in mind that cellular signaling cascades are closely intertwined with each other. For example, PTEN regulates the Src family: the combination of the Src family kinases inhibitor dasatinib and anti-CTLA-4 mAb reduces the number of MDSCs and Tregs, increases CD8^+^ T cells and thus results synergistic only in TGF-βR1/*PTEN*-loss mice [125,126]. A further example of crosstalk between PTEN and other pathways in regulating sensitivity/resistance to therapeutic immune checkpoint targeting is the interaction between PTEN expression and DNA mismatch repair (MMR): in a report focusing on PTEN immunohistochemistry (IHC) as a diagnostic tool for the screening of MMR-proficient breast cancer, PTEN-integrity seems to represent a protective factor against MMR deficiency in breast cancer [127].

Dynamic clonal tumor heterogeneity, often due to the selective pressure of specific drug treatments, further complicates the dissection of how the tumor genetic landscape influences immune homeostasis and response to immunotherapy. In that respect, a recent study on melanoma patients analyzed biopsies derived from patients who initially responded to either anti-PD-1 or anti-PD-1 plus anti-CTLA-4 mAb. The authors identified an association between *PTEN* mutations and patients who no longer respond to immunotherapy (28%). Among negative checkpoint modulators, V-domain Ig suppressor of T cell activation (VISTA) represents an interesting inhibitory immune-checkpoint protein, highly expressed on tumor-infiltrating leucocytes [128]. Resistance to anti-PD-1 was associated not only to decreased levels of the MHC components HLA-A and HLA-DPB1, but also to the accumulation of VISTA^+^ lymphocytes in biopsies taken at progression, as compared to pre-treatment biopsies [129]. The discrepancy between the percentage of PTEN-loss cases, as analyzed by IHC, and the high percentage of PD-L1 and VISTA^+^ lymphocytes (28% vs. 61% and 67%, respectively) in melanoma biopsies at progression suggests that the PTEN protein might be inactive, albeit present, in this context. More recently, Trujillo and his group carried out a similar analysis in order to define the characteristics of treatment-resistant metastases after combined anti-CTLA-4 plus anti-PD-1 therapy. This study shows that acquired immune resistance is mediated by: (i) T cell response insufficiency or tumor escape mechanisms (i.e., loss of antigens expression); (ii) overcoming of a specific memory-response against tumor cells, due to biallelic *PTEN* mutation and poor T cells infiltration [130]. In this regard, acquired immune resistance to anti PD-1 therapy is associated with changes in neoantigen presentation in the presence of biallelic *PTEN*-loss in patients affected by metastatic uterine leiomyosarcoma [131].

### 4.2. PTEN and Non-Conventional Immunotherapy

Even responses to the new immunotherapy approaches being developed (Table 1) might be influenced by the tumor genetic background, in particular by *PTEN* status, although data available are limited to in vitro and mice models [132]. One emerging approach exploits the use of OX40 agonist-based combinations, as investigated by Peng and collaborators. Based on results obtained in PTEN-loss/*BRAF*-mut melanoma, the authors demonstrated that a combination of OX40 receptor agonists and PI3K inhibitors successfully improves tumor immune response by promoting the proliferation and cytotoxicity of CD8^+^ T cells at the tumor site [133]. Similar results were also obtained by Kroon et al., who demonstrated that concomitant targeting of PD-1 and CD137 improves radiotherapy efficacy, in a *BRAF^V600E^*/PTEN-loss melanoma mouse model. This promising effect is mediated by an increased percentage of infiltrating CD8^+^ and CD4^+^ T cells, as demonstrated by high levels of granzyme B-expressing cells [134]. Such preclinical evidence could collectively represent a rationale for the treatment of PTEN-loss melanoma, highlighting the possibility to tailor treatment based on the best predicted response. An anti-TIM-3 mAb was investigated for its efficacy in TGF-βR/PTEN-loss head and neck squamous cell carcinoma transgenic mice. In the absence of treatment, these mice are characterized by both high percentage of TIM-3 Tregs and M2 macrophages; conversely, anti-TIM-3 mAb treatment significantly reduces the Tregs population but not M2 macrophages, hence enhancing the antitumor immune response [135]. 

CD20 represents a B cells marker expressed during B cells differentiation and rituximab, an anti-CD20 mAb approved by food and drug administration (FDA) in 1998, is used in cancer and autoimmune disease treatment for B cells depletion [136]. Gupta and collaborators demonstrated a link between PTEN and PI3K pathway activation and the response of human lymphoma cell lines to hexavalent antibodies (HexAbs) (anti-CD20, anti-CD22 and anti-CD22-20). HexAbs upregulate p38 and PTEN levels, leading to the activation of pro-apoptotic signals; moreover, the use of the PI3K inhibitor LY294002 increases apoptosis induced by HexAbs [137]. More recently, Baritaki and coworkers observed a similar response using the anti-CD20 mAb LFB-R603 in lymphoma cell lines: LFB-R603 inhibits both SNAIL and NF-κB and induces RKIP and PTEN, which in turn restores TRAIL-dependent apoptosis in TRAIL resistant non-Hodgin’s B-cell lymphoma [138].

Recent data in PTEN-loss mice developing head and neck tumors show that the upregulation of LAMTOR5 (a novel oncoprotein belonging to PI3K pathway) is associated with not only poor OS and lymph node grade, but also with F4/80 (a marker of mice macrophages) and VISTA levels. Consistently, the expression of VISTA as a T lymphocyte-activation inhibitor is associated with immunosuppressive microenvironment [139]. VISTA^+^ lymphocyte accumulation is also observed in melanoma patients with acquired resistance to immune checkpoint inhibitors and *PTEN* mutation. Such evidence could constitute the rationale of novel therapeutic strategies targeting VISTA in selected PTEN-loss melanoma patients [129].

## 5. Conclusions

Recent advances in immunotherapy (particularly the clinical use of immune checkpoint inhibitors) represent a turning point in the fight against cancer. However, mechanisms of immune escape and acquired resistance to immunotherapy hamper therapeutic efficacy resulting in treatment failure in a substantial proportion of patients. In that respect, certain genetic tumor characteristics, such as the lack of expression of the tumor suppressor PTEN, are of particular interest, due to their involvement in modulating both the tumor and the immune cell compartments of the TME towards an aggressive, therapy-resistant phenotype. Comprehensive PTEN status analysis could be crucial to refine the prediction of response to therapy in individual patients and to define new combinatorial approaches, in order to tailor the right therapies to patients who can actually benefit the most.

## Figures and Tables

**Figure 1 ijms-21-05337-f001:**
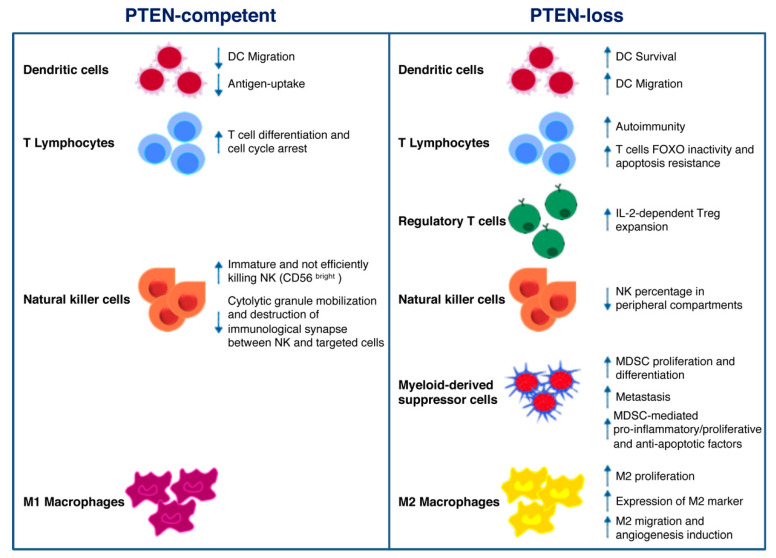
**Schematic illustration of phosphatase and tensin homolog on chromosome 10 (PTEN) function in immune cells.** PTEN modulates several microenvironmental stimuli and immune cells processes, according to the cell type in which it is expressed (**left panel**) or not. As for cancer cells, the lack of PTEN activity mainly correlates with immune escape mechanisms and protumoral immune cells infiltration/expansion (**right panel**).

**Figure 2 ijms-21-05337-f002:**
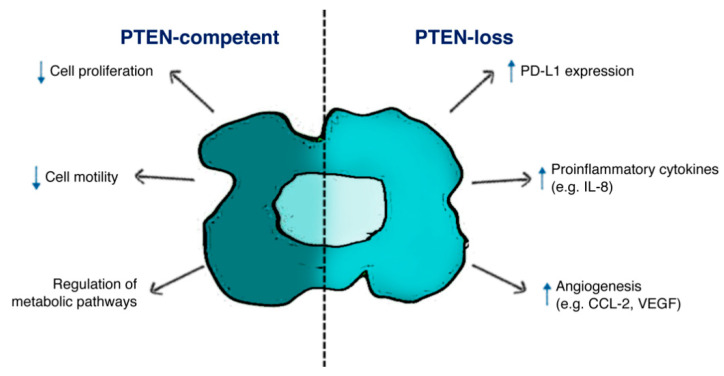
**Schematic illustration of PTEN function in tumor cells.** PTEN-status in cancer cells affects both biological features in cancer cells survival and tumor microenvironment (TME) composition, according to the expression of specific ligands (i.e., PD-L1) and soluble factors (i.e., IL-8, CCL-2, VEGF).

**Table 1 ijms-21-05337-t001:** Immunotherapy strategies against cancer.

Strategy		Biological Target(s)/Aim	Drug	Combination(s)	Effects on Tumor Growth	Ref
Effector mechanisms stimulation	Vaccine	DCs	Sipuleucel-T (2010)		−	[14]
		Tumor antigens			n.a.	[15,16]
		Nucleic acids			n.a.	[15,16]
	Oncolytic virus	Cancer cells lysis and increased immune response	Talimogene laherparepvec (T-VEC)		−	[17,18]
	Adoptive T cells	TILs			±	[19,20]
		CAR-T	Kymriah and Yescarta (2017)		−	[21]
Immunosuppressive mechanisms counteraction	Tregs inhibition	IL-2/CD25/chemotherapeutics	Cyclophosphamide		±	[22,23,24]
	Inhibitory checkpoint blockade	CTLA-4 PD-1/PD-L1	Ipilimumab plus nivolumab (2018)	CTLA-4 plus PD-1	−	[25]
			BMS-986207 plus nivolumab (NCT02913313)	II generation immunotherapy targets (LAG-3/TIM-3/TIGIT)	u.i.	[26,27]
				Costimulatory receptors (4-1BB)	±	[28,29,30]
				Conventional therapies	−	[31,32]
				Epigenetic modulators	u.i.	[33]

+, tumor progression; −, tumor regression; CAR-T, chimeric antigen receptor T cells; CTLA-4, cytotoxic T-lymphocyte antigen 4; DCs, dendritic cells; IL, interleukin; LAG, lymphocyte activation gene; n.a., not assessed; PD-1, programmed cell death-1; PD-L1, program death ligand; TIGIT, T cell immunoglobulin and ITIM domain; TILs, tumor-infiltrating lymphocytes; TIM, T-cell immunoglobulin mucin; Tregs, regulatory T cells; u.i., under investigation.

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
