# Peer review of "PTEN Function at the Interface between Cancer and Tumor Microenvironment: Implications for Response to Immunotherapy"

_ijms, 2020, doi:10.3390/ijms21155337_

Round 1

Reviewer 1 Report

The authors have addressed satisfactorily the issues raised by this reviewer

Author Response

Dear Editor,

enclosed please find the revised manuscript entitled “PTEN function at the interface between cancer and tumor microenvironment: implications for response to immunotherapy”, by Fabiana Conciatori and co-authors.

First, we would like to thank the reviewers for their helpful and constructive comments; we have tried our best to respond to all of them and believe that the revised manuscript has now substantially improved. We thus hope it can be considered for publication in the Special Issue entitled “Tumour Suppressor Function” in International Journal of Molecular Sciences.

Reviewer #1

We thank the reviewer for his/her recognition of the relevance of our report.

Reviewer #2

We thank the reviewer for his/her helpful comments/suggestions; in particular:

  • We thank the Reviewer for this comment. In this review, we aim only at highlight the role of PTEN mutations/expression in non-cancer cells in the tumor microenvironment. The role of PTEN in cancer cells is out the scope of this manuscript, hence we corrected the misunderstanding sentences, as appropriate.
  • We added “the” before TME as suggested.
  • We thank the Reviewer for the suggestion, and we broke some sentences in two smaller sentences, in order to help the readers.
  • We replaced “PTEN lies” with “PTEN activity” (new line #51), “miR” with “miRs” (new line #309), “to” with “with” (new line #745), “literature data” with “published data” (new line #864), “influence” with “influences” (new line #876), “granzime” with “granzyme” (new line #922); we deleted the first “to” in the line #207.
  • Table 1: we corrected the name of the two subgroups of treatments. We hope that they are now correctly and completely visible, and that the style of the table has not changed during the resubmission.
  • New lines #87-89: we rewrote the sentences.
  • We replaced “TIL”, “DC”, “APC”, “MDSC” and “NK” with “TILs”, “DCs”, “APCs”, “MDSCs” and “NKs”, as suggested.
  • New lines #304-306: we rewrote the sentences.
  • We used italics for all the gene names, as kindly suggested.
  • We corrected all the abbreviations through the text.
  • We modified the Figure 2, as kindly suggested.
  • PTEN-loss through the manuscript is referred as either PTEN gene alterations or completely lacking PTEN protein expression, as we published in Milella M. et al., Sci Rep 2017. In order to help the reader, we added a sentence in the Introduction section (line #34).
  • As suggested, we checked all the references.

All authors have read and approved the final version of the manuscript that is being submitted and all concur with the submission. On behalf of all authors, I also hereby certify that the present manuscript is not under consideration and will not be submitted elsewhere before completion of the review process.

Reviewer 2 Report

The authors should conscientiously address the below points for this manuscript to be deemed suitable for review.  

> The authors state in the abstract that the current knowledge on PTEN functions in different TME compartments (cancer, immune and stromal cells). However the section titles are only focussed on immune cells (3.1) and stromal cells (3.2). Where is the section on the cancer/tumour?

> The above comment is also not consistent with the different TME compartments specified in Line 57.

> In many uses of TME ‘the’ should be placed before for better readability (eg lines 22, 42, 81, 102, 107).

> Many sentences are long and convoluted these should be broken up into multiple sentences for clarity (eg 51-55, 211-214, 284-288,  312-315, 321-324, 328-332, 349-353)

> Line 44: PTEN lies? doesn’t make sense

> Table 1: Stimulate effectors mechanisms? doesn’t make sense

> Table 2: Counteract immunosuppressive? doesn’t make sense

> Rewrite Line 71-73, and make focus therapeutic vaccines

> Line 78, 79: TILs

> Lines 116 onward: check all uses of DC, APC and MDSC, in many cases plural should be used DCs, APCs & MDSCs

> Lines 178 & 179: clarify what PTEN-ko NK stage is?(x2)

> Line 183 onwards: italicise all uses of gene names for PTEN, miRs, FOG2, STAT3, Dicer1

> Line 185: microRNAs (miRs)

> Line 193: bone marrow-derived MDSCs

> Line 250: LOH is not defined.

> Figure 2: the gradient does not make sense. PTEN-competent should be darker blue, PTEN-loss (PTEN-mutants, see below comment) should be lighter blue. A dashed line halfway as a vertical axis should be used to divide this figure.

> The use of PTEN-loss throughout the manuscript is not used accurately or correctly in many cases. PTEN-mutant of PTEN-null (if known) would be better suited.

> Line 288: ‘.. with activation’

> Line 324: OS and TNM are not defined.

> Line 353: literature data?

> Line 364: influences

> Line 371: ‘associated to not only to’ ??

> LIne 395: ‘granzyme’

> Please check all references to ensure that they have been cited properly

Author Response

Dear Editor,

enclosed please find the revised manuscript entitled “PTEN function at the interface between cancer and tumor microenvironment: implications for response to immunotherapy”, by Fabiana Conciatori and co-authors.

First, we would like to thank the reviewers for their helpful and constructive comments; we have tried our best to respond to all of them and believe that the revised manuscript has now substantially improved. We thus hope it can be considered for publication in the Special Issue entitled “Tumour Suppressor Function” in International Journal of Molecular Sciences.

Reviewer #1

We thank the reviewer for his/her recognition of the relevance of our report.

Reviewer #2

We thank the reviewer for his/her helpful comments/suggestions; in particular:

  • We thank the Reviewer for this comment. In this review, we aim only at highlight the role of PTEN mutations/expression in non-cancer cells in the tumor microenvironment. The role of PTEN in cancer cells is out the scope of this manuscript, hence we corrected the misunderstanding sentences, as appropriate.
  • We added “the” before TME as suggested.
  • We thank the Reviewer for the suggestion, and we broke some sentences in two smaller sentences, in order to help the readers.
  • We replaced “PTEN lies” with “PTEN activity” (new line #51), “miR” with “miRs” (new line #309), “to” with “with” (new line #745), “literature data” with “published data” (new line #864), “influence” with “influences” (new line #876), “granzime” with “granzyme” (new line #922); we deleted the first “to” in the line #207.
  • Table 1: we corrected the name of the two subgroups of treatments. We hope that they are now correctly and completely visible, and that the style of the table has not changed during the resubmission.
  • New lines #87-89: we rewrote the sentences.
  • We replaced “TIL”, “DC”, “APC”, “MDSC” and “NK” with “TILs”, “DCs”, “APCs”, “MDSCs” and “NKs”, as suggested.
  • New lines #304-306: we rewrote the sentences.
  • We used italics for all the gene names, as kindly suggested.
  • We corrected all the abbreviations through the text.
  • We modified the Figure 2, as kindly suggested.
  • PTEN-loss through the manuscript is referred as either PTEN gene alterations or completely lacking PTEN protein expression, as we published in Milella M. et al., Sci Rep 2017. In order to help the reader, we added a sentence in the Introduction section (line #34).
  • As suggested, we checked all the references.

All authors have read and approved the final version of the manuscript that is being submitted and all concur with the submission. On behalf of all authors, I also hereby certify that the present manuscript is not under consideration and will not be submitted elsewhere before completion of the review process.

This manuscript is a resubmission of an earlier submission. The following is a list of the peer review reports and author responses from that submission.

Round 1

Reviewer 1 Report

The paper by Fabiana Conciatori et al. entitled “Cancer immunoevasion: the biology behind PTEN-loss role in immunotherapy resistance” performs a review of the recent literature on the links between structural and functional alterations of the tumor suppressor PTEN and immunogenicity induced by tumor processes. It is a detailed review of the literature, exhaustive regarding recent publications which finds interest because of the current development of immune checkpoint inhibitors for therapeutic purposes in tumor pathology.

I have two concerns

1/ The first part of the review entitled “immunotherapy” and devoted to all therapeutic approaches based on the strengthening of immune defenses which have been used in tumor pathology is quite far from the subject of this review which is focused on PTEN. It should be much more concise and reduced to a simple paragraph allowing to introduce the current issue of the links between PTEN and anti-tumor immunity.

2/ The review carried out in the 2nd part, entitled "PTEN in immunoevasion" is complete and seeks to collect all the publications published on the subject but an equal weight is given to cutting edge articles and to papers like the reference 158 (Lopez et al. Int. J. Mol. Sci. 2020) whose results, out of step with known data, probably suffer from significant methodological biases. A critical look must be taken on the publications and the most convincing results must be highlighted compared to the minor publications.

Author Response

Dear Editor,

First, we would like to thank the reviewer(s) for their helpful and constructive comments; we have tried our best to respond to all of them and believe that the revised manuscript has now substantially improved. We thus hope it can be considered for publication in the Special Issue entitled “Tumour Suppressor Function” in International Journal of Molecular Sciences.

As suggested by the Reviewer #1, we modified the manuscript as following:

  1. We agree with the Reviewer and, as suggested, the "Immunotherapy" paragraph is now more concise. We modified this section in order to present a general overview of the current knowledge about immunotherapy strategies (which are not only related to the well-known CAR-T and PD1/PD-L1 inhibition), by removing the specific molecular mechanisms that are not completely related to the topic of our manuscript (e.g. cancer vaccines and oncolytic viruses). Substantial modifications were made in “Stimulate the effector machineries” paragraph. As opposite, in order to help the reader in understanding the PTEN-related mechanisms in immunotherapy regulation (which is the focus of this review and reported in the last part of the manuscript), we maintained the “Inhibitory checkpoints blockade” section discussed in detail (e.g. PD-1 and PD-L1). Indeed, despite the promising results obtained by immune checkpoint inhibitors, many discrepancies are still present: identifying patients who can best respond to immunotherapy remains the main goal.
  2. We agree with the Reviewer on that we discussed more in detail some papers which suffers from methodological biases as compared to those of great impact, in the previous version of the manuscript. We have now reduced the comments on references #160 (Lopetz and coworkers), according to this suggestion. However, our aim was also to highlight that the methodological detection of PTEN analysis is fundamental to correctly classified patients and critically analyzed results from preclinical and clinical data. Indeed, the classification between “PTEN-competent” and “PTEN-loss” in cancer cells is a very oversimplification of the complex tumor dynamics, in addition to the bidirectional interactions between all the cells in TME (cancer, stromal and immune cells) which should be always considered. Hence, according to this goal, we focused our attention on giving to the reader tools for a critical interpretation of the data reported in the literature, which may seem controversial from a deep analysis.

As suggested by the Reviewer #2, we modified the manuscript as following:

  1. We thank the Reviewer for these comments and corrections. Indeed, in Figure 1 there was a mistake in positioning both PI3K and PTEN. Moreover, in the previous version of the figure we aimed to just simply describe that mTORC2 activation is mediated by PI3K signaling: indeed, the activation signaling downstream PIP3 formation is very complex and not related to just few proteins (as we represented in Conciatori F. et a., 201, Cancers, 10.3390/cancers10010023). We now changed the Figure 1, as suggested by the Reviewer, showing that mTORC2 activation is mediated by mTORC1 activation.
  2. We appreciate the Reviewer’s suggestion and we think that this new figure will help the reader to have a schematic view of PTEN role in both cancer and stromal cells. Indeed, an additional schematic Figure 3 was added in the text, in order to highlight the role of PTEN expression in cancer cells in affecting not only tumor features but also immunoescape mechanisms.
  3. We thank the Reviewer for suggesting us some typos mistakes and other correct references. All the grammatical inaccuracies are now corrected through the text and we added new references.

Yours sincerely,

Chiara Bazzichetto

Reviewer 2 Report

Conciatori et al. review the role of PTEN in the control of cancer immunoevasion and in the efficiency of current anticancer immunotherapies. The review is timely and well written, and will be of interest for cancer immunotherapy researchers and clinicians.

Some specific points:

  1. The schematic depiction in Fig. 1 is wrong and should be amended: the location of PTEN and PI3K should be shifted in the depiction. Due to this “typo”, it is not clear for this reviewer whether the authors wanted to put an arrow from PI3K directly targeting mTORC2. ¿It does exist a direct link (not mediated by PIP3) between PI3K and mTORC2? This should be revised accordingly.
  1. It is suggested to include an additional schematic figure (maybe as a Fig. 2) highlighting (as it is made in the text) the distinct functional role of PTEN, in the context of cancer immunoevasion and tumor progression, in both the tumor cells and in the immune and stromal cells.
  1. It is suggested to add some references that update the information on the addressed subjects:

- Line 36: it is suggested to update citations and include Lee et al. 2015 PMID 29858604 together with current reference 2

- Line 52: it is suggested to update citations and include Pulido et al. 2019 PMID 31501265 together with current references 3,4

- Line 125: it is suggested to update citations and include Song et al. 2018 PMID 29407608 together with current reference 17

- Line 343 (or somewhere else in this section): it is suggested to include Taylor et al. 2019 PMID 31501268, as a recent update on the role of PTEN in immune cells

  1. Other small corrections or suggestions

- Line 134: “European Medicines Agency”

- Line 220: Please check if you want to keep “Actually” at the beginning of the sentence. Maybe you were meaning “Currently”?

- Line 233: sentence a bit wrong? Please check and rewrite

- Line 242: delete “last month”

- Line 319: delete “a”

-Line 325: sentence very long and difficult to follow, please revise (“characterization”?)

- Line 350: please rephrase

- Line 452: same comment that for line 220 regarding “actually”

- Lines 771-775: very long sentence and with extra words: please rephrase

Author Response

(The authors gave the same response as above.)

Round 2

Reviewer 1 Report

The new version of the article "Cancer immunoevasion: the biology behind PTEN3 loss role in immunotherapy resistance" takes into account the remarks made by deleting a few paragraphs from the original text. The article remains long but documented. I have no additional comments to do.